# Differential Roles of Neural Integrity, Physical Activity and Depression in Frailty: Sex-Related Differences

**DOI:** 10.3390/brainsci13060950

**Published:** 2023-06-14

**Authors:** Sara Isernia, Marta Cazzoli, Gisella Baglio, Monia Cabinio, Federica Rossetto, Fabrizio Giunco, Francesca Baglio, Valeria Blasi

**Affiliations:** IRCCS Fondazione Don Carlo Gnocchi ONLUS, 20148 Milan, Italy; sisernia@dongnocchi.it (S.I.); mcazzoli@dongnocchi.it (M.C.); gbaglio@dongnocchi.it (G.B.); mcabinio@dongnocchi.it (M.C.); frossetto@dongnocchi.it (F.R.); fgiunco@dongnocchi.it (F.G.); vblasi@dongnocchi.it (V.B.)

**Keywords:** frailty, ageing, MRI, brain, sex, gender, depression, physical therapy

## Abstract

The frailty sex paradox has recently gained attention. At all ages, females are more likely to be frail and show a more severe phenotype but have a higher survival rate compared to males. The main aim was to test sex-specific differences in frailty syndrome using a multimodal evaluation from clinical and imaging data to deepen the understanding of different underlying mechanisms involved in the two sexes, and thus understand the association with different risk factors. Ninety-six community-dwelling older adults were characterized by clinical underpinnings (Fried’s frailty indicators: comorbidity, depression, global cognitive level, physical activity, autonomy), and neural integrity (T1-weighted brain 3T MRI). The frailty × sex interaction in clinical and neural profiles was tested. Additionally, frailty risk factors were identified in the two sexes separately. Results showed that fragility was associated with an increment of depressive symptomatology in females, while a decrement in physical activity was observed already in the pre-frail stage in males. Finally, different risk factors were observed in the two groups: significant frailty predictors were neural integrity and physical activity in males, and age and depression in females. These data support the starting hypothesis of at least partially different mechanisms involved in the frailty phenotype between men and women.

## 1. Introduction

Frailty is a complex phenotype characterized by physiological reserve decay and stressor susceptibility. Frailty syndrome currently represents a global public health priority, given its increasing incidence linked to the longevity increment, and related significant care needs, adverse outcomes, and disability risk [1].

Evidence of sex-specific differences in frailty syndrome have recently gained attention [2,3,4,5,6]. At all ages, females are more likely to be frail and show a more severe phenotype compared to males. Paradoxically, women present a higher survival rate though. In fact, when compared to males of the same age, females present a worse health status, suggesting higher frailty, whilst also an superior susceptibility to death, suggesting less frailty [4]. This has been called the sex-paradox phenomenon in the frailty phenotype. Although the causes and underlying mechanisms of the sex-related differences in frailty are yet to be fully understood, the combined effect of biological, behavioral, and social factors has been implicated [4,5]. Therefore, knowing how to further understand the differential roles of clinical and behavioral risk factors between women and men in the fragility phenotype may be crucial.

Several studies have tried to address this issue in relation to the “chronic disease hypothesis” claiming the differential incidence of life-threatening diseases between women and men. Especially, females’ frailty phenotype may peculiarly reflect mood alteration, such as depression symptomatology [7,8], and cognitive impairment [9], together with the accumulation of non-fatal chronic disease [2]. Instead, males are likely to show physical health weakness involving more life-threatening diseases such as heart failure and pulmonary diseases [2,4]. Nevertheless, data from longitudinal studies also lend support to the hypothesis of qualitative and not only quantitative sex differences in chronic medical conditions [10], suggesting greater physiological reserve in females providing greater resilience to frailty [11].

Despite the abundance of studies investigating the frailty paradox, the role of depression and motor activity has not been clarified. Moreover, several factors that might be implicated still need to be explored. One such factor is the potential role of the neural component, which has been largely neglected so far. The neural integrity component could add relevant information, especially if connected to the behavioral component such as cognitive impairment and mood. Neural integrity, indeed, could be related to the recent observation that frailty may increase the risk of future cognitive decline, and vice versa, cognitive impairment may represent a risk factor for frailty [12,13]. Such evidence suggests a complex interplay between cognition and frailty in aging, and led to the conceptualization of cognitive frailty [14]. In line with these studies, neural integrity could represent a relevant factor in explaining the risk of developing a frailty phenotype and addressing differences in resilience/risk factors between women and men.

The aim of this study was, thus, to investigate the differential predictive role between women and men of clinical, behavioral, and neural indices in the frailty phenotype. The starting hypothesis was that the frailty phenotype could be the expression of different underlying mechanisms involved in the two sexes and thus associated with different risk factors. With this aim, we investigated a cohort of 96 community-dwelling older adults with a comprehensive clinical battery and a high-resolution structural brain MRI.

## 2. Materials and Methods

In this study, we tested sex differences in the frailty phenotype from data collected within the “Transcription factor PREP1 in the frailty phenotype” project (https://www.ifom.eu/bandoCariplo/, (accessed on 13 June 2023)).

### 2.1. Participants

Subjects included in the analysis (N = 96) were selected from the “Transcription factor PREP1 in the frailty phenotype” database (N = 116). This database included neural and clinical data on a cohort of community-dwelling older adults recruited at the IRCCS Santa Maria Nascente and the Palazzolo Institute of the Don Gnocchi Foundation of Milan. In detail, participants enrolled in the project fulfilled the following inclusion criteria: age ≥65; absence of a diagnosis of dementia (Mini-Mental State Examination score >18); absence of unstable acute or chronic conditions and unstable pharmacological treatment; absence of Parkinson’s disease, multiple sclerosis, or infectious diseases; and have read and signed the informed consent module approved by the Don Gnocchi Foundation Ethics Committee.

Subjects included in the present study were selected according to the presence of the following data:-Fried’s frailty phenotype: based on five frailty indicators: unintentional weight loss in the prior year (≥4.5 kg), grip weakness measured by a manual dynamometer (20% below the norm, based on gender and BMI), perceived exhaustion in their daily routine, slowness in walking (20% below the norm, based on gender and height) measured by the 10 Meter Walking Test [15], and low physical activity (20% below the norm) in terms of kcal/week measured by Minnesota Leisure Time Activity Questionnaire [16]. Based on Fried’s instructions [17], people presenting three or more frailty indicators were classified as frail, those who reported one or two indicators as prefrail, and people without any indicators classed as robust.-Demographic characteristics: age, sex, and years of education and anamnestic information collected during the clinical interview, such as current and remote comorbidities, surgical interventions, and pharmacological treatments.-Motor activities: the Physical Activity Scale for the elderly (PASE, [18]) questionnaire. The PASE score ranges from 0 to 793, with a higher score indicating a greater physical activity routine.-Cognitive level: the Montreal Cognitive Assessment (MoCA, [19]). The MoCA total score ranges from 0 to 30, with a higher score suggesting a greater global cognitive level.-Psycho-Behavioral level: the Center for Epidemiologic Studies Depression Scale (CES-D, [20]), for a measure of the frequency of depression-related symptomatology. The scale score ranges from 0 to 60, with a higher score indicating greater depressive symptoms.-Participation in daily life: the Alzheimer’s Disease Cooperative Study—Activities of Daily Living Scale (ADCS-ADL, [21]). The ADCS-ADL score ranges from 0 to 78, with a higher score suggesting a greater autonomy level.-Neural integrity: a 3T MRI brain examination including a T1-3D magnetization-prepared rapid acquisition gradient echo (MPRAGE, 0.80 mm^3^, TR/TE: 2300/3.1 ms, FOV: 256 × 240 mm) sequence for brain morphology evaluation, fluid-attenuated inversion recovery (FLAIR) (0.4 × 0.4 × 1 mm^3^, TR/TE: 5000/394 ms, FOV: 256 × 230 mm) for white matter hyper-intensities, and T2-weighted sequence to exclude gross brain abnormalities.

### 2.2. Materials

In the present study, comorbidity and neural indexes were computed.

The classical Charlson Comorbidity Index (CCI, [22,23]) was calculated based on the anamnestic information collected during the clinical interview. A higher CCI indicates a greater mortality risk and comorbid condition.

Additionally, to compute a global morphological neural index (Neural Index), white matter hyperintensities were manually segmented to derive T1-3D lesion-filled images, which were analyzed in Freesurfer software (v. 6.0, https://surfer.nmr.mgh.harvard.edu/ (accessed on 13 June 2023)) According to Klapwijk et al. [24], the images were manually corrected, when necessary. Then, the total gray volume, the total cerebral white matter volume, and the estimated intracranial volume were extracted according to Fischl et al. [25]. Total gray volume and total cerebral white matter volume were summed and normalized to the estimated intracranial volume to compute the neural index. The neural index ranged from 0 to 1, with a higher index indicating a greater global neural integrity.

### 2.3. Statistical Analysis

The Statistical Package for Social Sciences (SPSS, IBM Corporation, v. 28) was used for the statistical analysis.

Skewness, kurtosis, and histogram plots were visually explored to check the variables’ normal distributions and perform parametric or non-parametric analyses when adequate. Mean, standard deviations, and frequencies were reported to describe the participants’ characteristics. An independent t-test and chi-squared χ^2^ test were run to compare male and female groups’ demographics and frailty indicators. An ANCOVA (age inserted as a covariate) was used to test the effect of the frailty phenotype (robust, prefrail, and frail group) and sex (males and females) on clinical and neuropsychological profiles. Bonferroni’s post hoc test was run to interpret statistically significant results. A partial correlation (age inserted as a covariate) was run to check statistically significant associations between frailty score, defined as the total number of frailty indicators detected (score 0 to 5), and clinical, neuropsychological, and neural index variables separately for male and female groups. Finally, demographical, clinical, neuropsychological, and neural variables were inserted in a multiple regression model to test the best predictors of the frailty score separately for the male and female groups.

All the statistical analysis models were two-tailed. Results were considered statistically significant when the *p*-value was <0.05.

## 3. Results

### 3.1. Participants Characteristics

In total, ninety-six subjects (58 females; mean age = 75.49 ± 6.62; mean education = 11.29 ± 3.85) were included in the analysis. The mean frailty score in the whole group was 1.22 ± 1.17. No differences between males and females were registered for age and education, but females showed a statistically significantly higher frailty score than males (*p* = 0.007, Cohen’s d = 0.548). Males presented more pulmonary comorbidities than females (*p* = 0.008), whilst women were more likely to present hypertension condition than men (*p* = 0.014) (Table 1).

### 3.2. Sex Differences in Fried’s Frailty Indicators

Table 2 reports the results of the Chi squares χ^2^ analyses on sex distribution for each frail indicator in the whole, pre-frail, and frail groups. In the whole group, people presenting exhaustion, slowness, and hand-grip weakness were significantly more likely to be females (*p* < 0.05). In the pre-frail group, significantly more females than males showed hand-grip weakness (*p* < 0.05). In the frail group, females were more likely to show body-weight loss and slowness (*p* < 0.05) than males (*p* < 0.06). Instead, men were more likely to show hand-grip weakness (*p* < 0.05) and tended to be more likely to show exhaustion and low physical activity (*p* < 0.06) than females.

### 3.3. Effects of Sex and Frailty on Clinical, Psychobehavioral and Neural Profile

Table 3 reports the results of the ANCOVA (age inserted as a covariate) to test the effect of the Frailty phenotype (robust, prefrail, and frail group) and Sex (males and females) on clinical, behavioral, and neural integrity profiles.

Results showed a significant effect of frailty phenotype in the comparison between robust, prefrail, and frail groups for CES-D (*p* < 0.001, partial η^2^ = 0.199, ω = 0.990), MoCA (*p* = 0.040, partial η^2^ = 0.070, ω = 0.617), and PASE variables (*p* = 0.008, partial η^2^ = 0.103, ω = 0.808), while no effect was detected for education, ADCS, and neural index. Especially frail people reported more depressive symptoms than the pre-frail and robust groups, and pre-frail showed more depressive symptoms than the Robust group. The frail group reported a lower MoCA score than the robust group. Moreover, the robust group showed a higher PASE score than the frail and pre-frail groups.

Moreover, a significant effect of sex was detected for education (*p* = 0.034, partial η^2^ = 0.049, ω = 0.567), CES-D (*p* < 0.001, partial η^2^ = 0.148, ω = 0.973) and the neural index (*p* = 0.031, partial η^2^ = 0.051, ω = 0.582). Specifically, according to the Bonferroni post hoc analysis, males were likely to report a higher level of education than females, females were more likely to show depressive symptoms, and males showed a higher neural index than females.

Finally, a significant interaction between the frailty phenotype and sex was observed for the CES-D (*p* = 0.014, partial η^2^ = 0.091, ω = 0.750) and the PASE (*p* = 0.042, partial η^2^ = 0.069, ω = 0.612) (Figure 1. In detail, in the female group, the worsening of the fragility condition was associated with an increment in depressive symptomatology while in the male group the depressive symptoms remained mostly stable across all frailty phenotypes. For the PASE variable, in males, the decrement in physical activity is observed already in the pre-frail stage while in the females this occurs mainly at the frail stage.

Explorative correlations between frailty score, and comorbidities (CCI), motor, cognitive, participation, and neural integrity level revealed a different pattern of associations in males and females. In males, a significant association of frailty was observed with the PASE (r = −0.326, *p* = 0.049) and neural index (r = −0.344, *p* = 0.037), while in females, frailty correlated with CES-D (r = 0.728, *p* < 0.001), and MoCA (r = −0.303, *p* = 0.022).

Table 4 reports the multiple regression model to test the best predictors of frailty separately for males and females. In the male group, neural index, and PASE score were significant predictors of frailty (R^2^ = 0.356, Figure 2). Differently, in the female group, age and CES-D score were the only predictors of frailty (R^2^ = 0.723, Figure 3).

## 4. Discussion

The differential predictive role of demographical, cognitive, participation, motor, and neural domains in the frailty phenotype were investigated in a sex-stratified cohort of community-dwelling older adults. In line with the literature [2,3,4,5], in our cohort, women were frailer and had a different fragility profile than men. Thus, our cohort can be considered representative of the general population in terms of frailty prevalence in the two sexes. Interestingly, when we focused on frailty indicators, males and females showed distinct profiles: fragility in women was characterized by the prevalence of unintentional weight loss and slowness, while men had a prevalence of hand-grip strength loss, exhaustion, and low physical activity.

When investigating factors related to frailty phenotype in the two sexes, unexpectantly, the demographic variables were not implicated. In fact, the level of education and comorbidities, as calculated with CCI, had no role on the frailty phenotype, and on how the frailty phenotype was expressed in the two sexes. It has to be noted that in the classical CCI, life-threatening chronic conditions are mainly considered, while other chronic comorbidities, such as hypertension, do not weigh on the index. Differences between males and females, though, were detected in the type of comorbidities observed in our cohort. Specifically, women showed a higher prevalence of hypertension while men had a higher occurrence of pulmonary disorders. Taken together, these results support the chronic disease hypothesis claiming a greater female prevalence of ‘non-life-threatening’ diseases, such as hypertension, whereas males had a higher prevalence of ‘life-threatening’ diseases [4,7]. Another area that does not appear to explain the distinct patterns of fragility among men and women is the level of participation in everyday life. This result should be interpreted in association with the lack of any effect of participation on the frailty phenotype. Our participants were selected among community-dwelling people, and it is likely that autonomy in daily living was preserved for all. Instead, considering the role of the cognitive domain (MoCA test) on frailty phenotype in males and females, we found a significant association, but not a predictive relation, of the cognitive domain with the level of fragility in women. This is in line with results from a recent systematic review and meta-analysis showing how the frailty phenotype is linked with all dementia conditions among community-dwelling older people, and that women with frailty had a higher risk of Alzheimer’s disease than men with frailty. However, in our cohort, the global cognitive level was not a factor implicated in the frailty phenotype neither in men nor in women. However, it should be mentioned that this finding could be linked to a selection bias in our cohort. In our population, indeed, according to the inclusion criteria, people with a low/very low global cognitive level could not be considered for the present study.

It is nevertheless interesting to note that the psycho-behavioral (CES-D scale) and motor (PASE) domains were likely to have a role in explaining the distinct frailty phenotype profile in males and females. Our data showed a straightforward link between depression and fragility in women but not in men. The link between the Frailty phenotype and depression was explained by the interaction between the Frailty score and sex. Indeed, our data showed that the worsening of the fragility condition was associated with an increment in depressive symptomatology only in women. Differently, men showed instead rather stable depressive symptomatology across all frailty phenotypes. Moreover, depression was also significantly correlated with frailty score, and most importantly, was also only one of the main risk-predicting factors in women. The co-existence of frailty and depression in the frailty condition is amply documented in the literature, with the prevalence of depression in frail older adults at 46.5% [26,27]. As of today, only a few contributions supported a higher prevalence of depression in females, plausibly exacerbating disability related to comorbidities [8,28,29]. As stated by Park and Ko [2], whether depression can increase the risk of frailty in a sex-specific manner is still currently unclear. The herein-presented data support this hypothesis of an increased level of depression being a risk factor for the frailty phenotype only in women. Considering the motor domain, the significant interaction between frailty score and sex found for the PASE scale highlights how the reduction in physical activity in men occurs in the early pre-frailty stage while in women this happens mainly at the frailty stage. Moreover, the PASE score correlated with the frailty score only in men. Finally, physical activity was one of the two predicting factors of the risk of fragility in men. Interestingly, even if robust males reported higher physical activity, plausibly related to a better attitude than females toward motor exercises [30], already in the pre-frail stage the physical activities appeared strongly affected in males. This result may propose sex-specific frailty underlying mechanisms that mostly involve physical aspects in men. Our data are in line with results from a study involving 6803 adults aged 25–64 years from the Belgian National Health Interview Survey [30] investigating the differential impact of physical activity on mental health between men and women. Data from this survey showed how physical activity had a positive effect on mental health in both sexes, but in men, this was especially true for high-intensity physical activity while in women for mild-intensity activity.

One of the most relevant findings of this study is the link between the neural integrity index with fragility in men. Indeed, the neural domain constituted a relevant factor for sex-related differences in the frailty phenotype, as highlighted in both the correlation and the linear regression analyses. To our knowledge, this is the first time that neural integrity was explored to differentiate its role in the risk of developing frailty syndrome between men and women. Age-related gross anatomical brain changes are widely known [31] in senescence, linked to cognitive decline manifestations, especially impacting episodic memory and working memory, and attention. Neural integrity and physical activity though are tightly linked, as demonstrated by the beneficial effects of physical activity on several conditions [32], such as cardiovascular, metabolic, and neurologic disorders, including dementia, and sarcopenia. In our cohort, the relation between neural integrity and frailty phenotype in men plausibly indicates that neural decay is tightly linked to physical fragility in this group. Instead, the absence of correlation between frailty score and neural integrity in females may be related to other epigenetic and environmental factors, such as reserve and compensation mechanisms plausibly attenuating the effects of neural decay on frailty phenotype. The link between physical activity and neural integrity is documented by several studies showing the role of physical activity in the prevention of dementia [33,34], and the association of physical activity with lower age-related gray and white matter loss and with lower levels of neurotoxic factors [35]. The mechanisms involved in the role of physical activity in protecting brain health and promoting neuroplasticity include the improvement in respiratory function [36], cerebral perfusion, and reduction in the risk of vascular diseases (e.g., plaque deposits in arteries, atherosclerosis, hypertension, and stroke) [37,38,39,40].

This study is not exempt from limitations. First, this is a cross-sectional investigation, and the findings need to be confirmed by a further longitudinal study. Moreover, our cohort was small in size, and the sub-groups (frail, pre-frail, robust group) included a small number of subjects. Nevertheless, the subjects included in this analysis were well-characterized in terms of clinical, behavioral, and also neural integrity profiles.

## 5. Conclusions

In conclusion, our findings hint how critical it is to deepen understanding of the differential role of neural, clinical and behavioral risk factors between women and men in the frailty phenotype. This is a relevant issue considering the potential reversibility of the frailty phenotype. Taken together, the results suggest the importance of sex-specific prevention and treatment strategies for frailty. On the one hand, females could benefit from screening, diagnosis, and interventions to counteract depression [5]. Among interventions, social enhancement activities may positively impact mood in women at risk of frailty. Music-based interventions, such as dance-based group therapy, for example, can be effective in the prevention and treatment of depression through the stimulation of music-evoked emotions and pleasure that stimulate the reward and emotion network [41] and through the stimulation of physical activity itself [42] to prevent frailty conditions, and reduced neural integrity may be an informative index of an increased risk of frailty syndrome in men. Indeed, physical activity has been shown to stimulate the growth of brain-derived neurotrophic factor (BDNF) and insulin-like growth factor-1 [35,43]. The down-regulation of oxidative stress and inflammatory responses [39,44], and the reduction in neurotoxic factors such as beta-amyloid and excessive glucose [37,45] have also been implicated. Longitudinal evaluation of our cohort and other studies with longitudinal designs may help to better understand gender trajectories among frailty aging.

## Figures and Tables

**Figure 1 brainsci-13-00950-f001:**
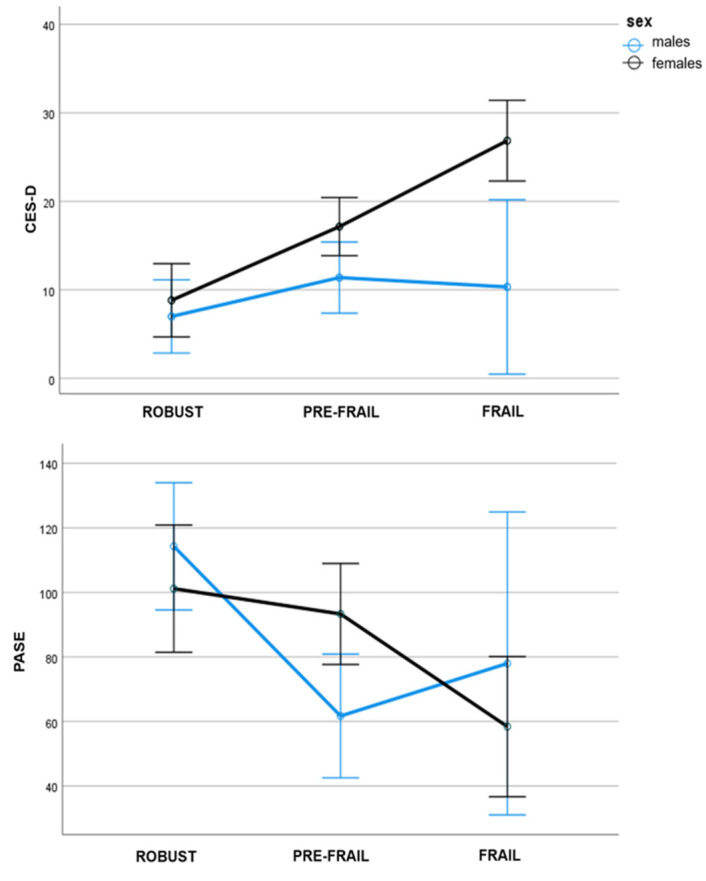
Sex × Frailty interaction in CES-D and PASE score.

**Figure 2 brainsci-13-00950-f002:**
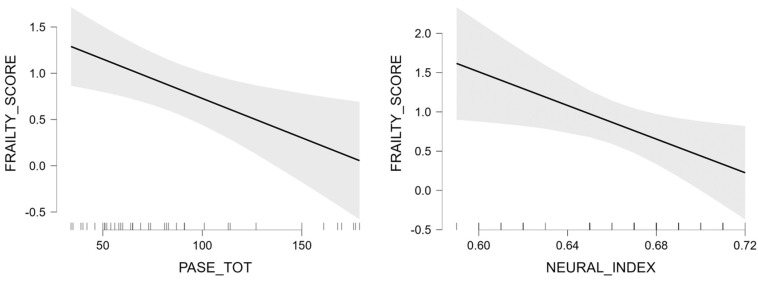
Marginal effect plots of the effects of PASE, and neural index on frailty score in the males’ group.

**Figure 3 brainsci-13-00950-f003:**
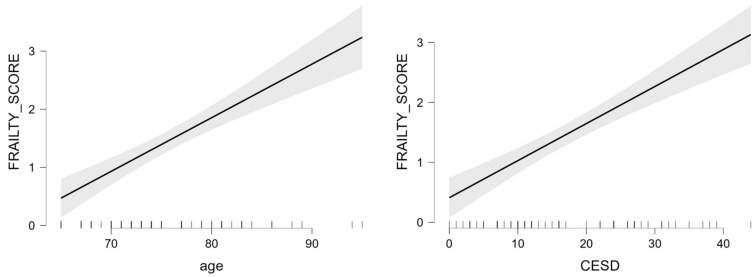
Marginal effect plots of the effects of age and CES-D on frailty score in the female group.

**Table 1 brainsci-13-00950-t001:** Participants’ characteristics and males’ vs. females’ comparison results.

	All	Males	Females	Males-Females Comparison (*p*)
N	96	38	58	
Age, M (sd)	75.49 (6.62)	75.05 (6.08)	75.78 (7.00)	*t* = −0.52 (0.603)
Education, M (sd)	11.29 (3.85)	12.13 (3.73)	10.74 (3.86)	*t* = 1.75 (0.084)
Frailty score, M (sd)	1.22 (1.17)	0.84 (0.97)	1.47 (1.23)	*t* = −2.76 (0.007)
Comorbidities				
Myocardial infarction, %	10.42	13.16	8.62	χ^2^ = 0.38 (0.535)
heart failure, %	7.29	7.89	6.90	χ^2^ = 0.03 (0.855)
* lung disease, %	21.87	39.47	10.34	χ^2^ = 11.33 (0.008)
Diabetes, %	5.20	10.53	1.72	χ^2^ = 2.96 (0.085)
moderate-severe renal disease, %	1.04	2.63	0.00	χ^2^ = 1.74 (0.187)
chronic liver disease, %	2.08	2.63	1.72	χ^2^ = 0.10 (0.755)
gastric peptic ulcer, %	1.04	0.00	1.72	χ^2^ = 0.76 (0.383)
Cancer, %	6.25	10.53	3.45	χ^2^ = 2.05 (0.152)
rheumatic/connective tissue disease, %	1.04	0.00	1.72	χ^2^ = 0.76 (0.383)
Hypertension, %	60.00	47.37	72.41	χ^2^ = 6.03 (0.014)
Charlson Comorbidity Index ^§^, M (sd)	0.60 (0.91)	0.45 (0.72)	0.71 (1.01)	*t* = −1.37 (0.087)

Legend: N, number; M, mean; sd, standard deviation; * 19 Chronic obstructive pulmonary disease; 1 sleep apnea treated with Continuous positive airway pressure; 1 asthma. ^§^ = does not include hypertension.

**Table 2 brainsci-13-00950-t002:** Male and female distribution of Fried’s frailty indicators and chi-squared results.

	All	Robust	Pre-Frail	Frail
Total [N]	96	34	45	17
M/F	38/58	17/17	18/27	3/14
Body-weight loss				
Total [N;(%)]	13; (13%)		5; (11%)	8; (47%)
M/F [N; (%)]	3/10; (7%/17%)	-	2/3; (11%/11%)	1/7; (33%/50%)
χ^2^ (*p*)	3.77 (0.052)		0.20 (0.655)	4.50 (0.034) *
Exhaustion				
Total [N;(%)]	24; (25%)		11; (24%)	13; (76%)
M/F [N;(%)]	7/17; (18%/29%)	-	4/7; (22%/26%)	3/10; (100%/71%)
χ^2^ (*p*)	4.17 (0.041) *		0.82 (0.366)	3.77 (0.052)
Low physical activity				
Total [N;(%)]	20; (2)		10; (22)	10; (59)
M/F [N;(%)]	7/14; (18%/24%)	-	4/6; (22%/22%)	2/8; (67%/57%)
χ^2^ (*p*)	2.33(0.127)		0.40 (0.527)	3.60 (0.058)
Slowness				
Total [N;(%)]	14; (15)		6; (13)	8; (47)
M/F [N; (%)]	1/13; (3%/22%)	-	1/5; (5%/18%)	0/8; (0%/57%)
χ^2^ (*p*)	10.29 (0.001) **		2.67 (0.102)	- ***
Hand-grip strength				
Total [N;(%)]	43; (45)		29; (64)	14; (82)
M/F [N;(%)]	12/31; (32%/53%)	-	9/20; (50%/74%)	3/11; (100%/79%)
χ^2^ (*p*)	8.39 (0.004) **		4.17 (0.041) *	4.57 (0.033) *

Legend: F, females; M, males; N, number; ***, < 0.001; **, < 0.01; *, < 0.05.

**Table 3 brainsci-13-00950-t003:** Behavioral profile of groups and ANCOVA results (age as a covariate).

	Robust		Prefrail		Frail		ANCOVA			
	Males (n = 17)	Females (n = 17)	Males (n = 18)	Females (n = 27)	Males (n = 3)	Females (n = 14)	Sex F	Frailty F	Sex × Frailty F	Bonferroni’s Post Hoc
Demographics										
Education, M (sd)	11.71 (3.84)	12.41 (2.78)	12.00 (3.69)	10.26 (4.17)	15.33 (2.52)	9.64 (3.95)	4.63 *	0.76	2.27	Males > Females
CCI, M (sd)	0.53 (0.72)	0.47 (0.72)	0.33 (0.68)	0.85 (1.10)	0.67 (1.15)	0.71 (0.85)	0.41	0.11	0.36	-
Motor Activities										
PASE, M (sd)	114.29 (46.39)	101.18 (48.69)	61.72 (31.19)	93.33 (42.45)	78.00 (20.07)	58.43 (32.95)	0.00	5.08 **	3.30 *	Robust > Frail, Prefrail
Cognitive Level										
MoCA, M (sd)	22.37 (2.01)	23.44 (2.53)	21.82 (3.14)	21.15 (2.80)	19.91 (3.11)	20.38 (4.62)	0.15	3.33 *	0.80	Frail < Robust
Psycho-Behavioral Level										
CES-D, M (sd)	7.00 (5.97)	8.82 (7.10)	11.39 (8.94)	17.15 (10.04)	10.33 (4.51)	26.86 (9.78)	15.45 ***	11.05 ***	4.44 *	Females > Males; Frail > Pre-Frail, Robust; PreFrail > Robust
Participation in daily life										
ADCS, M (sd)	77.18 (1.51)	77.00 (1.37)	73.00 (8.32)	72.44 (9.93)	78.00 (0.00)	66.21 (17.35)	1.97	0.72	0.39	-
Neural integrity										
Neural Index, M (sd)	0.66 (0.03)	0.68 (0.06)	0.66 (0.03)	0.68 (0.088)	0.63 (0.03)	0.66 (0.04)	4.80 *	0.77	0.87	Males > Females

Legend: ADCS, Alzheimer’s Disease Cooperative Study—Activities of Daily Living Scale; CCI, Charlson Comorbidity Index; CES-D, Center for Epidemiologic Studies Depression Scale; M, mean; MoCA, Montreal Cognitive Assessment; PASE, Physical Activity Scale for Elderly; sd, standard deviation. *, *p* < 0.05; **, *p* < 0.01; ***, *p* < 0.001.

**Table 4 brainsci-13-00950-t004:** Significant frailty predictors in male and female groups.

	Predictors	Standardized β	SE	*t*	Targeted *p*	F	Omnibus *p*	R^2^
Males	age	0.00	0.02	0.13	0.897	4.56	0.005	0.356
	education	0.07	0.04	2.00	0.054			
	PASE	−0.01	0.00	−2.79	0.009			
	neural index	−11.06	4.56	−2.43	0.021			
	Intercept	7.75	4.37	1.77	0.085			
Females	age	0.09	0.01	6.94	<0.001	34.60	<0.001	0.723
	education	0.01	0.03	0.56	0.579			
	CES-D	0.62	0.01	7.48	<0.001			
	MoCA	−0.05	0.03	−1.86	0.068			
	Intercept	−5.55	1.25	−4.44	<0.001			

Legend: CES-D, Center for Epidemiologic Studies Depression Scale; MoCA, Montreal Cognitive Assessment; PASE, Physical Activity Scale for Elderly.

## Data Availability

The raw data supporting the conclusions of this article will be made available upon reasonable request by the corresponding author.

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
