# Peer review of "Differential Roles of Neural Integrity, Physical Activity and Depression in Frailty: Sex-Related Differences"

_brainsci, 2023, doi:10.3390/brainsci13060950_

Round 1
Reviewer 1 Report
I have read your study and found some typing errors (ex. line 64:replace the word << the>> with the word this or our study).
Very good work
Very good quality of language but authors should check the text for typing errors.
Author Response
We thank the reviewer for the comments.
-We checked the manuscript for typing errors and corrected the text as suggested (see red text).
Reviewer 2 Report
I congratulate the authors for taking up such an important topic. I have only minor comments to consider.
1. 1 Introduction
I thing that worth to add in introduction:
The prevalence of FS in patients with various clinical conditions (e.g. CVD) and its proven impact on the occurrence of many adverse medical consequences should therefore be associated with the systematic identification of FS in clinical practice. Frailty syndrome potentially is a reversible condition, so there is a need to establish appropriate management strategies (also includinc sex differences), as well as to adopt appropriate definitions and tools for identification of FS in clinical pratice.
Citation: https://www.mdpi.com/1660-4601/19/4/2234
2. 2 The methodology is not in doubt.
3. 3 The results are presented correctly.
4. 4 The discussion is interesting and includes current literature.
5. 5 The conclusions are supported by the results.
no comments
Author Response
We thank the reviewer for the suggestions.
1. We added in the conclusions section the concept of reversibility of the frailty syndrome (see red text).
Reviewer 3 Report
The paper is interesting, when comparing the differences in certain variables, in older people, in terms of frailty.
Obtains curious results that can be contrasted with other studies.
The statistics part is correct with the data you have and what you are looking for.
Some considerations are:
The conclusions must be more humble, and put words as they suggest, not demonstrate. The sample is very small, the study is cross-sectional, and the subsamples by various criteria are minimal.
Regarding the tables, indicate if the given value is a percentage or a score, as it is not clear. That is, in table 1, it points to do a chi, or a t, I understand that for the chi they are percentages, and for the t test, the mean is used, but it is not clear.
You should point out more characteristics of the sample, they are people who live alone, they are in a nursing home. etc
Author Response
We thank the reviewer for the suggestions.
1- We softened our conclusions (see red text). We agree with the reviewer; our study has limitations. We reported them in the limitation section of the discussion (see red text).
2- In line with the suggestions of the reviewer, we modified Table 1 to clarify which data are percentages and which statistical test we run for the comparison (t-test or chi-squared) (see Table 1, red text).
3- We specified in the “Participants” section of the Method, that all participants were community-dwelling people (see red text).